# Perceptions of Knowledge and Experience in Nature-Based Health Interventions

**DOI:** 10.3390/ijerph21091182

**Published:** 2024-09-05

**Authors:** Carissa R. Smock, Courtney L. Schultz, Jeanette Gustat, Robby Layton, Sandy J. Slater

**Affiliations:** 1Department of Leadership, Management & Human Capital, School of Business and Economics, National University, San Diego, CA 92123, USA; 2Health & Technology Partners, Milwaukee, WI 53202, USA; courtney@healthandtechnologypartners.com; 3Department of Epidemiology, School of Public Health and Tropical Medicine, Tulane University, New Orleans, LA 70112, USA; gustat@tulane.edu; 4ActivEnviro, Longmont, CO 80504, USA; rob@dcla.net; 5School of Pharmacy, Bachelor of Science in Public Health Program, Concordia University Wisconsin, Mequon, WI 53097, USA; sandra.slater@cuw.edu

**Keywords:** physical activity, nature, interventions, public health, health services

## Abstract

Although perceptions and uses vary, nature-based health interventions (NBHIs) help facilitate the additional health benefits of physical activity (PA) experienced in nature, thereby reducing all-cause morbidity and mortality. The purpose of this mixed-methods, cross-sectional study was to better understand perceptions, terminology, and participation in NBHIs. A questionnaire was developed by reviewing validated instruments and gauging expert experience with stakeholders. Distributed electronically, a community partner listserv promoting active living served as the population. Quantitative questionnaire measures assessed familiarity with NBHI terms, concepts, experience, how NBHI should be used, and the importance of NBHI components. Qualitative themes included the strengths and weaknesses of NBHIs. Participants (*n* = 53) were familiar with the terms nature play (82%), forest bathing (78%), and park and nature prescriptions (74%) and moderately familiar with NBHIs (5-point Likert scale, M = 3.27, SD = 1.17). Most thought NBHIs could be useful in treating physical (96%) and mental health conditions (100%) and would follow or write one (80%). The location was reported as the most important component, followed by access, social comfort, dosage, and ongoing support. This study suggests stakeholders are familiar with and support NBHIs conceptually; however, policies, support, and funding opportunities are needed to operationalize components to increase use of NBHIs.

## 1. Introduction

It is clear that exposure to nature provides important and vast benefits from vitamin D absorption, additional immune response, and stress reduction [1,2] to mediation, socializing, and physical activity [3,4]. These benefits can lead to a sense of life satisfaction and happiness due to mood improvement, anxiety reduction, social connection, sense of place, sense of community, and active living [5]. Nature-based health interventions (NBHIs) are programs, activities, or strategies that aim to engage people in nature–based experiences with the specific goal of achieving improved health and well-being [4,5]. Additionally, NBHIs have been shown to further reduce costs on healthcare systems [2,5]; therefore, there are multiple benefits of increasing and innovating NBHIs [6].

### 1.1. Importance of Physical Activity in Nature

Exposure to nature and the outdoors has positive mental, physical, and social health benefits including increased physical activity (PA) [7,8,9,10]. Physical inactivity is a public health problem globally as 1 in 4 adults 18 years and older do not meet the globally recommended levels of physical activity (PA) [3], including at least 150 min of PA per week [10]. These recommendations are due to the dose–response relationship between PA and health benefits. Adults who acquire at least 150 min of moderate intensity or 75 min of vigorous intensity PA per week reduce mortality by 15% across their lifespan and a 26% mortality reduction can be achieved by acquiring at least 300 min of PA per week. Associated healthcare costs are also reduced [10].

Adult PA has increased from 18% of adults in 2008 to 24% of adults in 2018 [11]; however, about 75% of adults are still not meeting the PA guideline [12]. A large number of the adults who do not meet PA recommendations report that this is due to inequities in access to safe natural environments for PA [13]. Engaging in PA in nature provides additional benefits from being outdoors. These benefits include vitamin D absorption, additional immune response, and stress reduction [1,3,14]. Increasingly, healthcare providers are directing their patients to spend time and participate in activities at parks and other outdoor places by giving them a referral or prescription to spend time in nature—one form of a nature-based health intervention.

The socio-ecological model (SEM) includes domains that can be used to address the impact of NBHIs at each level including individual, social/intrapersonal, community/environment (built and natural), and policy, and it is recommended to guide complex challenges in access to nature for PA. Therefore, SEM guides this study across each of the socio-ecological domains needed to increase promotion of NBHIs [4,5].

### 1.2. Study Purpose

There are varying perceptions about NBHIs, uses of different terminology, and opinions about how stakeholders participate with NBHIs [3]. Additionally, the measurement of and operationalization of these interventions is often limited [4]. It is unclear if an operationalized framework or measures could be understood among the many fields of stakeholders. Additionally, the roles these stakeholders are open to serving in and currently participating in is uncertain. The purpose of this mixed methods, cross-sectional study was to better understand perceptions, terminology, and participation in NBHIs to guide future research priorities.

## 2. Materials and Methods

This pilot study was designed using Creswell and Plano Clark’s (2018) [15] mixed-methods convergent parallel design to simultaneously collect quantitative and qualitative data using a cross-sectional questionnaire design (see Figure 1).

Step 1 utilized content expert researchers and practitioners to review validated instruments including measures of familiarity with NBHI terms, concepts, experience, how NBHI should be used, and the importance of components. Step 2 included concurrently collecting quantitative data and qualitative data from a convenience sample for the purpose of pilot testing the NBHI instrument. Step 3 included analyzing descriptive statistics, and Step 4 involved qualitative analysis including open coding of themes. Step 5 served the purpose of triangulation, converging, and corroborating the study findings through comparing the qualitative and quantitative data [16].

### 2.1. Step 1 Instrument Creation

To create the questionnaire, step 1 included a network of public health researchers and practitioners that support local-, state-, and national-level policy approaches to influence PA opportunities. We adapted questionnaire questions related to PA counseling practices, barriers, and resources as needed. We also created and/or adapted questionnaire questions that combined familiarity with NBHIs and beliefs about their use and components [17,18,19,20,21]. The questionnaire consisted of 22 items, mostly fixed-response Likert scale items with three free-response questions to give respondents the opportunity to provide further detail. The average duration to complete the instrument was 5–7 min.

Five categories were identified to form the basis for the assessment instrument. (1) Familiarity with NBHI terms and concepts was assessed by asking participants to select all NBHI terms they were familiar with from a provided list and enter any additional term(s) they use for NBHIs. Familiarity with NBHIs was measured by defining NBHIs and then asking participants to rate their familiarity from extremely familiar to not familiar on a 5-point Likert scale. (2) Experience with NBHIs was measured by listing NBHI training, administration, and participation in NBHIs and asking participants to select all that applied. (3) Use of NBHIs included nine items participants were asked to answer about the context of their NBHI use on a 3-point Likert scale including agree, disagree, and unsure. (4) Importance of components of NBHIs was measured using five descriptions of components, which participants were asked to rate on a 5-point Likert scale from not important at all to very important with an option to fill in additional component(s) that should be included. (5) Strengths and weaknesses of NBHIs were assessed in two separate open-ended fields. Categories 1–4 focused on quantitative assessment and category 5 included qualitative assessment. Additionally, an open-ended field was provided for participants to share any other comments or thoughts about NBHIs. The only demographic information collected was the occupational field.

Following the initial instrument adoption and construction, the Delphi method [22] was used with five experts in the field to finalize the instrument. The experts represent a mix of practitioners and researchers. The refined instrument was then re-administered to the same five experts for field testing to assess the clarity of items, derive a time estimate for questionnaire completion, provide face validity, and identify potential content gaps. Based on comments from the field group, debriefing meetings were held with the experts to further refine instrument language and assess the time for completion. Pilot participants agreed that 5–7 min was an appropriate estimate to share with participants. The final version of the questionnaire consisted of 22 items.

### 2.2. Step 2: Data Collection

A convenience sample of adults employed or volunteering in recreation, land management, planning, or health were recruited from a community partner organization listserv of emails (ActivEnviro, formerly GP RED, www.activenviro.org, emailed on: 25 October 2021). Qualtrics’ online questionnaire tool was used to distribute the questionnaire link via the community partner’s email listserv of more than 15,000 members. A reminder email was sent after one month. Consent was obtained from participants on the first screen of the questionnaire in Qualtrics and was required for participants to continue to the questionnaire. No personally identifying information was requested from participants. The National University Institutional Review Board approved the questionnaire and recruitment process; this study’s authors are all members of the Physical Activity Policy Research and Evaluation Network (PAPREN) Parks and Green Space workgroup and represent both practitioners and academic researchers. PAPREN is a national network of the CDC focused on advancing the evidence base to support PA policy.

### 2.3. Step 3: Quantitative Data Analysis

All statistical analyses were conducted with SPSS version 28.01.01. Variables were derived from the 22-item questionnaire, available in its entirety through request to the first author. Descriptive analyses were conducted to summarize familiarity with NBHIs, beliefs about use and components of NBHIs, and participant characteristics. Means and standard deviations were calculated for the Likert-type questions and frequencies were determined for the categorical responses.

### 2.4. Step 4: Qualitative Data Analysis

Qualitative data were analyzed using Saldaña’s (2016) [23] evaluative and descriptive coding, including applying codes to qualitative data to assign judgment about the merit, worth, or significance of familiarity with NBHIs and beliefs about their use and components. We identified codes from the open-ended questions by identifying exemplary text and then identifying the parent code and associated subcodes using Microsoft Excel. The main themes and subthemes were discussed among all study team members.

### 2.5. Step 5: Integrating Quantitative and Qualitative Data

The fifth step integrated the data to compare quantitative beliefs about NBHIs to qualitative strengths and quantitative components of NBHIs to qualitative weaknesses. Through triangulation and convergence of the data, this approach developed a more complete picture of the applicability and usability of NBHIs [15,16]. This allowed the study team to compare, corroborate, and examine facets of the quantitative and qualitative data. Therefore, the data were assessed using parallel constructs for both types of data and the results were compared through transforming the qualitative data set into quantitative scores through ranking and jointly displaying both forms of data [15]. The two types of data provide validation for each other and create a foundation for deriving decisions about changes to the items in the NBHI tool while providing triangulation for perceptions about combined familiarity with NBHIs and beliefs about their use and components.

## 3. Results

### 3.1. Quantitative Results

Fifty-three individuals participated in the pilot study. The majority of respondents (62%) worked in the parks and recreation field, with other fields of land management (25%), public health (23%), and others represented (Table 1). Many respondents were extremely or very familiar with NBHIs (34%), 25% were moderately familiar, and 26% were only slightly familiar. Only 4% of respondents were not at all familiar with NBHIs.

There are many terms used to name NBHIs. Respondents were asked to identify all the terms with which they were familiar. Of the 16 various terms for NBHIs presented, participants were most familiar with “nature play/wild play (77%), “forest bathing” (75%), and “park prescription” (70%) (Table 1).

While most participants were at least somewhat familiar with NBHIs, their familiarity came from reading about or attending seminars on NBHIs, rather than personally receiving or prescribing them. Table 1 shows that 77% of respondents had read about or attended a seminar on NBHIs. Some had prescribed (17%) or received (8%) them and 13% indicated they had no experience with NBHIs.

There was consistent agreement on several of the respondents’ beliefs about NBHIs. Every respondent agreed that NBHIs could be useful in treating mental health conditions (100%). Nearly all respondents agreed that NBHIs could be useful in treating physical health conditions (96%). Most disagreed that NBHIs should only be written by medical practitioners (68%), the NBHIs were not “real” medicine (82%), or that more research was needed (58%). Respondents were divided, however, on agreement over whether “NBHIs should be used for preventative care rather than treatment for existing conditions” with 34% agreeing, 38% disagreeing, and 26% unsure (Table 2).

There was also agreement on the importance of each of the components of NBHIs when using or prescribing them; over 80% of respondents indicated that each component was moderately to very important. The NBHI components assessed included dosage information or how much/often NBHIs should be used, location resources such as maps and directions, counseling or support such as text messaging and calls, access assistance including entrance fees, transportation and disability assistance, and social comfort, which communicates a sense of welcoming (Figure 2).

### 3.2. Qualitative Results

The qualitative analysis included thematic analysis of open-ended questionnaire responses about the strengths and weaknesses of NBHIs/prescriptions. The strengths of NBHIs resulted in six themes: (1) convenience and accessibility, (2) comprehensive well-being approach, (3) safety and minimal side effects, (4) public health awareness, (5) natural environment interplay, and (6) opportunities for partnerships. Table 3 includes subthemes and example quotes for each theme.

Respondents identified several advantages of NBHIs, highlighting their affordability, accessibility, and diverse health benefits. NBHIs were noted by respondents as holistic methods for enhancing overall well-being, addressing both physical and mental health concerns. A key highlight from the respondents was their emphasis that NBHIs could be easily administrated and potentially foster enduring lifestyle changes. Notably, respondents described NBHIs’ non-intrusive nature as particularly appealing to individuals wary of medication, while also being capable of promoting self-reliance, exploration, and social interaction. Respondents praised NBHIs for their dual role in addressing existing health issues and preventing chronic conditions; several noted that such interventions offer a drug-free, minimally impactful alternative. Several respondents mentioned that time spent outdoors has minimal side effects, which sets NBHIs apart from conventional medications. Additionally, nature-based prescriptions were described by respondents as important for facilitating personal connections with the natural world. Respondents also highlighted that NBHIs foster social cohesion and can be tailored to individual preferences and locales, cultivating positive habits and societal awareness of the benefits of outdoor activity. Overall, respondents’ comments underscore the multifaceted strengths of nature-based interventions, encompassing physical and mental health benefits, affordability, accessibility, and societal well-being.

Thematic analysis of qualitative, open-ended questionnaire responses about weaknesses of NBHIs/prescriptions resulted in five themes including (1) trust and acceptance, (2) support and resources, (3) implementation challenges, (4) access and accessibility, and (5) environmental concerns. Table 4 includes subthemes and example quotes for each theme. Respondents suggested that the weaknesses of NBHIs/prescriptions encompass various challenges. Respondents identified limited institutional support and healthcare provider education as key contributors to implementation difficulties. A lack of insurance coverage and workplace support, according to respondents, further impedes access. Additionally respondents pointed out that mainstream acceptance and credibility remain lacking, along with inadequate data collection and research on dosage. Additionally, respondents raised the concern that there is a lack of public understanding and trust in prescribers, often resulting in generic prescriptions without tailored approaches. Respondents also noted weakness of NBHIs as related to potential patients including equity issues, discomfort in outdoor settings, and the need for sustained engagement add to the complexities. Several respondents commented on accessibility barriers, including cost and transportation issues, which are perceived to hinder widespread adoption. Safety concerns, such as exposure to environmental hazards, also were seen as challenges of NBHIs by respondents. Overall, addressing the multifaceted weaknesses identified by respondents requires collaborative efforts to enhance the awareness, support, and integration of nature-based interventions into healthcare systems and public parks and greenspaces.

### 3.3. Qualitative and Quantitative Results

Data integration comparing quantitative to qualitative data revealed parallel constructs. Table 5 includes qualitative data transformed into quantitative scores resulting in ranking and jointly displaying both forms of data. Comparing the four highest ranking quantitative beliefs to qualitative strengths of NBHIs, *treating mental health conditions* had a higher ranking than the emergence of the subtheme of *mental health treatment* under the *comprehensive well-being approach*. Conversely, the quantitative construct of *treating health conditions* ranked lower than the emergence of this subtheme under the *comprehensive well-being approach*. The quantitative construct *promoting social benefits* ranked higher than the aligning subthemes under the *comprehensive well-being approach* and *natural environment interplay,* and the agreement with *following or writing a NBHI* was lower than the aligning subtheme theme under *convenience and accessibility*.

Table 6 includes comparing the quantitative components to qualitative weaknesses of NBHIs, quantitative measures *of location/resources* and *access assistance* both aligned as the two highest ranking with the emergence of the subthemes under *access and accessibility*. The quantitative measure *social comfort* was the second highest ranking, with the emergence of the aligning subtheme under *environmental concerns* ranking third. While *dosage information* ranked fourth in the quantitative construct, the aligning subtheme under *implementation challenges* ranked fifth. Finally, the quantitative construct of *counseling/support* ranked lowest, while the aligning subtheme under *implementation challenges* ranked fourth.

## 4. Discussion

The purpose of this study was to better understand the perceptions, terminology, and participation in NBHIs to support research. Aligning with the literature [13,17,21,24], the findings suggest that other NBHI stakeholders understand the importance of and are aware of NBHI but need support, including operationalized guidelines and training. This presents an opportunity for stakeholders across multiple sectors to leverage partnerships to operationalize guidelines and training to better understand how to increase access to NBHIs in all communities to improve overall population health and health outcomes.

### 4.1. Validation of Our Instrument

The questionnaire used for this study was developed and adapted by a network of public health researchers and practitioners, as we were unable to locate a questionnaire that would answer our research questions and serve the purpose of our study—to better understand perceptions, terminology, and participation in NBHIs. This network consists of academic researchers and practitioners that are members of the PAPREN Parks and Green Space workgroup supporting local-, state-, and national-level policy approaches to influence PA opportunities. The questionnaire questions were adapted from previously validated instruments when available [17,18,19,20,21], related to PA counseling practices, barriers, and resources needed. The remaining questionnaire questions that combined familiarity with NBHIs and beliefs about their use and components were created. The resulting questionnaire consisted of 22 items including mostly fixed-response Likert scale items with categories 1–4 identified to focus on a quantitative assessment of NBHI familiarity, experience, use, and perception of the importance of components, with category 5 including a qualitative assessment of the remaining ideas, strengths, and weaknesses of NBHIs.

Utilizing the Delphi method [21] with five experts in the field allowed for the instrument to be further refined. Additionally, field testing was conducted with these experts to ensure and assess the clarity of items, select a time estimate for questionnaire completion, determine face validity, and identify content gaps.

### 4.2. Implementation Barriers and Opportunities at Each SEM Level

Specifically, at the individual level of the SEM, those administering NBHIs need support to reduce barriers in administering NBHIs, including time, follow-up, and training. For those participating in NBHIs, proximate, accessible, safe, and familiar locations for PA increase adherence [17,20].

At the interpersonal level, referrals to nature pose challenges systematically in assigned roles and legal access to data [20,21,22,23,24]. At the interpersonal and community levels, NBHIs do not occur frequently due to a lack of access, training, and safety. Stakeholders across several industries, including healthcare, social services, local transit authorities, insurance, business, parks and recreation, and community advocates, have the potential to work together to test frameworks that identify NBHI opportunities, link NBHIs to clinical outcomes, and provide the scheduling and follow-up needed for increased uptake. Consistent with the findings from previous research [14,25], collaboration across industries will be critical in overcoming several of the implementation barriers identified in this study. Providers need additional training to overcome their educational gaps regarding prescribing NBHIs, but there is also a need for parks and greenspaces to improve inclusivity and accessibility so that people of all abilities and all baseline environmental knowledge can safely and confidently spend time outdoors. At the community level, some stakeholders are working together to offer equitable PA and nature programming; however, sustainability and limited offerings pose challenges [26].

At the policy level, health insurance plans designed to utilize a national care management model and the National Committee for Quality Assurance (NCQA) established a Population Health Management (PHM) strategy to assist in the management and resulting cost control of chronic diseases, improved wellness, and patient safety [27]. This includes community resources with the potential for the facilitation of NBHIs to help overcome these challenges [27].

These findings suggest NBHIs might be best served to focus on how stakeholders can bridge the gap between using nature or time spent outdoors as a healthcare modality and billable coding and reimbursable activities recognized by insurance providers. National care management models are being implemented as a way to drive and improve healthcare outcomes, reduce costs, and improve disease management. These care models could be an opportunity for care managers to utilize and track park referrals and nature prescriptions through implementing a PHM strategy [28].

Therefore, there is increased opportunity for organizations to reimagine and create sustainable, safe-access, operationalized NBHIs [22,29,30,31,32]. A community coalition of leaders from diverse sectors in health, conservation, urban design/planning, and park and recreation could leverage the opportunity to catalyze NBHIs to combat disparities in access to nature for PA. Therefore, NBHIs intersect multiple SEM domains and are needed to support calls to action for stakeholders to support access to NBHIs. The continued growth of NBHIs requires an ongoing cultural shift in the United States about the importance of health and nature and a commitment to cultivating spaces that benefit everyone.

### 4.3. Future Research Directions

Our use of the term NBHI in this study does not differentiate between nature prescriptions or nature programming [33]. Tailored approaches are required to create lasting health behavior change such as increased PA outdoors. A more nuanced exploration of these two complementary approaches is important to understand the continuum of opportunity that exists between them to increase PA during time in nature.

The strengths of NBHIs are compatible with existing health systems, recreation systems, and participants’ needs. Innovative solutions are needed to overcome the weakness of NBHIs that were identified in this study. To understand what is needed to remove barriers, we, therefore, recommend the development of a framework utilizing each level of SEM to address policy, training, and workforce capacity that engages communities as well as practitioners.

Much work is needed to better understand what the operationalized framework we recommend might look like that addresses access, participation barriers, and cultural perceptions between diverse populations at each level of the SME. It is suggested to focus on human–nature as a two-way, beneficial relationship that might reduce barriers and improve sustainability [33]. However, is often left out in health and conservation policies [33]. One suggested framework model that might uncover these pathways and capture some of the differences between cultures and populations is “A time with e-Natureza” (e-Nature), introduced by Leão et al. (2023) [34]. This framework focuses on the underlying interactions of nature-based health interventions including (1) esthetic and emotional experience; (2) multisensory integration experience; (3) knowledge experience; and (4) engagement experience. It seems this model would need further development to fully consider climate-related health risks, such as heatwaves or pollution, though it seems the “multisensory integration experience” may be used to consider this area. Therefore, limited information is understood about the contribution to climate resilience and environmental sustainability (Barragan-Jason et al., 2023) and current research is also unclear if specific types of nature settings are more beneficial for certain health outcomes. The use of validated instruments such as NatureScore to measure the quality of the setting and scales to measure attitudes [35], self-efficacy, and intentions about spending time in nature [36] may assist in capturing the more comprehensive benefits of nature on health [37].

### 4.4. Strengths and Limitations

The strengths of this study include the mixed methods design, providing both quantitative and qualitative results to examine perceptions, common terminology, and participation in NBHIs by a diverse group of stakeholders using validated questions, when available. However, some limitations should also be noted. First, the participants represent a convenience sample, limiting our ability to generalize findings to broader populations. However, participants did represent diverse industries (e.g., parks, recreation, preventive health, land management, education, and research). Second, while medical professionals participated in the study, they represented a small percentage of overall respondents. Future research should administer this questionnaire to a larger, more representative sample of stakeholders involved in NBHIs. Third, the questionnaire should be further validated, as not all questions used were previously available.

## 5. Conclusions

This study suggests stakeholders are familiar with and support encouraging access to nature through NBHIs. Policies, support programs, and funding opportunities may consider operationalizing components needed within communities to encourage the use of NBHIs. Rigorous evaluation is also needed to determine what combination of components is most effective for supporting adherence to NBHIs and improving long-term health outcomes.

Respondents to this study advocated for a holistic approach to NBHIs, incorporating education, community involvement, and systemic changes in healthcare and societal attitudes towards nature. NBHIs have enormous potential to improve both individual well-being and environmental health. However, achieving health reciprocity between people and nature requires sustained effort and collaboration across industries to overcome existing barriers. There exists a complexity around NBHI implementation that requires systemic change within healthcare, but it also requires increased public awareness and support. Collaborative efforts between parks, health agencies, and community organizations generate innovative programs that offer support, encouragement, and education, thereby addressing barriers such as safety concerns and a lack of familiarity with outdoor activities. Such efforts also require inclusivity; NBHIs need to cater to individuals of all abilities and disabilities, which will require advances in park accessibility.

## Figures and Tables

**Figure 1 ijerph-21-01182-f001:**
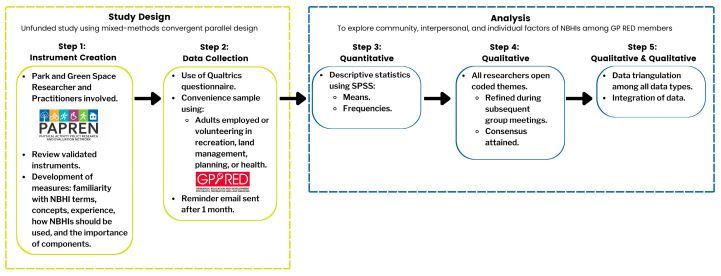
Pictorial representation of study design.

**Figure 2 ijerph-21-01182-f002:**
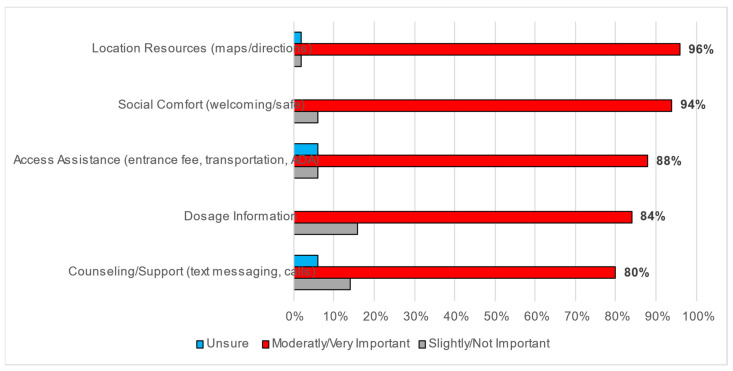
Importance of NBHI components when using or prescribing.

**Table 1 ijerph-21-01182-t001:** Participant demographics (n = 53) and nature-based heath intervention (NBHI) familiarity.

Characteristic	n	%/Mean
**Occupational field (%)**		
Parks and recreation	33	62
Land/trails management	13	25
Public health	12	23
Other	9	17
Non-profit organization	6	11
Research	7	13
Urban planning	6	11
Built environment	5	9
Community organization	5	9
Landscape architecture	5	9
Allied health	3	6
Medical	1	2
**Familiarity with terms (%)**		
Nature play/wild play	41	77
Forest bathing	40	75
Park prescription	37	70
Wilderness therapy	31	58
Walking prescription	29	55
Eco-therapy	28	53
Nature-based health interventions	26	49
Horticulture/gardening prescription	23	43
Green gyms/outdoor exercise groups	23	43
Outdoor prescription	21	40
Generic counseling to go outdoors	20	37
Trail prescription	19	36
Green exercise	18	34
Open-space prescription	14	26
Blue prescription	8	15
Other	5	9
**Experience with NBHIs (%)**		
Received NBHIs	4	8
Prescribed NBHIs	9	17
Friends/family received NBHIs	8	15
Read articles/attend seminars on NBHIs	41	77
No experience with NBHIs	7	13
Don’t know/unsure	2	4
**Familiarity with NBHIs (%)**		
Not familiar at all	2	4
Slightly familiar	14	26
Moderately familiar	13	25
Very familiar	13	25
Extremely familiar	10	19

**Table 2 ijerph-21-01182-t002:** Respondent agreement regarding beliefs about nature-based health interventions.

Questionnaire Statement	Agree	Disagree	Unsure
Should only be written by licensed medical practitioners	8%	68%	22%
Can be useful in treating physical health conditions	96%	0%	4%
Can be useful in treating mental health conditions	100%	-	-
Can be useful in promoting social benefits (cohesion, sense of place, and inclusion)	94%	2%	4%
Should be prescribed as preventative care rather than treatment for existing conditions	34%	38%	26%
Not “real” medicine	6%	82%	10%
Need more research before using in clinical practice	6%	58%	32%
Need more research to study side effects or negative impacts	24%	44%	30%
Based on current science, I would follow and/or write	80%	2%	16%

**Table 3 ijerph-21-01182-t003:** Quotes and themes: strengths of nature-based health interventions/prescriptions.

Questionnaire Question: What Are the Strengths of Nature-Based Health Interventions/Prescriptions?	
Theme/Subthemes	Quotation Example
Convenience and accessibility Availability Proximity Low cost Ease of administering Fun/enjoyable Customized	*“Low-cost, “fun” way to address health issues—whether mental or physical. Can be easily tailored to the individual, has variety of benefits beyond what a prescription might be written for (*e.g., *social benefits in addition to getting exercise)”.—General industry*
Comprehensive well-being approach Holistic well-being approach Lifestyle change promotion Socialization and connectedness Prevention and treatment Positive habit formation Mental health treatment	*“NBHI speaks to the whole person—physically, mentally, emotionally and socially”* *“Any prescription from a qualified professional can help increase time outdoors. The health benefits are clear from the research”.*
Safety and minimal side effects Absence of negative side effects Non-toxic Natural remedy	*“Unlike most medicines—no adverse side effects; “getting back to nature” can help mitigate the effects of modern life (crowds, tech, stress, pollution)”*
Public health awareness Public health benefit	*“Increased recognition and appreciation for open spaces/nature and their importance as a component of public health”.*
Natural environment interplay Awareness of nature Increased/diverse use Climate adaption Appreciation/enjoyment	*”Increased number of visitors, increased diversity of visitors, and increased awareness of nature spaces”.*
Opportunities for partnerships Strengthen connections Funding	*“Abundant opportunities at all scales—local, county, state, federal, private”.*

**Table 4 ijerph-21-01182-t004:** Reported weaknesses of nature-based health interventions/prescriptions.

Questionnaire Question: What Are the Weaknesses of Nature-Based Health Interventions/Prescriptions?	
Theme/Subthemes	Quotation Example
Trust and Acceptance Lack of understanding Lack of confidence Lack of familiarity Lack of credibility	*“The general public doesn’t understand the incredible importance of time outdoors. There are also issues of trust with prescribers”.* *“It is a newer science and people seem to be weary from the idea of it.* *Not enough medical authorities are participating yet”.*
Support and Resources Insurance support gap Lack of workplace support Institutional backing gap	*“Doctors are reluctant to prescribe for a variety of reasons—especially if they do not have positive personal outdoor experience.* *“Not backed by insurance. Expenses should be reimbursable”.* *“Not backed by workplace. How can people take the “medicine prescribed” if the workplace does not support the prescription?”*
Implementation Challenges Lack of follow-up mechanisms Lack of evaluation Lack of standard protocol Provider education gap Generic prescriptions	*“There are no randomized controlled trials (although one is currently in progress at Unity Healthcare in DC). Doctors get no training on the benefits of outdoor visits. Research on dosage is not sufficient to connect outdoor settings with the treatment of specific medical conditions. Doctors who are willing to prescribe have difficulty finding the time in the clinical setting”.* *“Healthcare providers are overwhelmed and adding this to their plate is challenging… clinicians want to have a way to measure follow-through and effectiveness, the data/process is not there yet”.*
Access and Accessibility Cost and transportation barriers Inequitable access Time constraints Limited capability of recipient Social/cultural issues	*“May not work if perceived barriers still exist, such as not feeling welcome or safe in the park/space/facility or if someone feels they can’t participate without the “right” clothing, shoes, equipment,* etc. *May be costly and hard to sustain without financial support to cover fees, transportation, program staff,* etc.*”.* *“Can have barriers to experiences-cost, transportation, not feeling safe, not accessible”*
Environmental Concerns Physical injury/accidents Allergens Environmental hazards Safety issues	*“Nature can have its own dangers-swimming in the ocean, hiking in the woods or high grass; even gardening could cause allergies for some individuals”* *“May not be easy to do in bad weather”* *“Wildlife can be intimidating”*

**Table 5 ijerph-21-01182-t005:** Qualitative and quantitative joint table: ranking comparison of beliefs and strengths.

Quantitative Construct Ranking—Beliefs	Qualitative Theme	Qualitative Rank
1. Can be useful in treating mental health conditions	Comprehensive well-being approach	2nd
2. Can be useful in treating health conditions	Comprehensive well-being approach	1st
3. Can be useful in promoting social benefits (cohesion, sense of place, and inclusion)	Comprehensive well-being approach Natural Environment Interplay	4th
4. Based on current science, I would follow and/or write	Convenience and accessibility	3rd

**Table 6 ijerph-21-01182-t006:** Qualitative and quantitative joint table: ranking comparison of components and weaknesses.

Quantitative Construct Components Weakness Ranking	Qualitative Weakness Theme	Qualitative Rank
1. Location/resources	Access and accessibility	1st
2. Social comfort	Environmental concerns	3rd
3. Access assistance	Access and accessibility	2nd
4. Dosage information	Implementation challenges	5th
5. Counseling/support	Implementation challenges	4th

## Data Availability

The aggregate data presented in this study are available on request from the corresponding author due to restrictions from the Institutional Review Board Statement to protect human subjects’ identity.

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
