# Peer review of "Perceptions of Knowledge and Experience in Nature-Based Health Interventions"

_ijerph, 2024, doi:10.3390/ijerph21091182_

Round 1
Reviewer 1 Report
Comments and Suggestions for Authors
I thank the authors and editor for the opportunity to review this manuscript. It is very well written and covers an important, under-researched topic. There are some minor edits that I have suggested, mainly concerning the Introduction section. Once these are addressed, I think the manuscript will be suitable for publishing and will be a nice addition to the literature.
Abstract:
I would add an ‘and’ between ‘(74%)’ and ‘moderately’ and delete the semicolon separating these two clauses in the following sentence: “Qualitative themes 18 included strengths and weaknesses of NBHI. Participants (n= 53) were familiar with terms nature 19 play (82%), forest bathing (78%), park and nature prescriptions (74%); moderately familiar with 20 NBHI (5-point Likert scale, M = 3.27, SD = 1.17).”
Introduction:
Currently, the Introduction section goes straight into section 1.1, which clearly has a strong focus on physical activity in nature. I think that the Introduction could be strengthened by having a brief section for 1.0 before going into 1.1. In this additional part, this is where I would set the scene with what we know about the importance of exposure to nature and the different experiences (e.g., mediation, socialising, physical activity) that can lead to broader health benefits, such as stress reduction, mood improvement, anxiety reduction, social connection, sense of place, sense of community, active living. Then, from there you can go into the specific focus on physical activity. The reason for this suggested change is that the title of the article gives the impression that the focus is on the range of health benefits of NBHI, but right from the start, the Introduction mainly just focuses on physical activity in nature. Further, on page 2 of the Introduction, there is the following sentence, “NBHI are programs, activities, or strategies that aim to engage people in nature–based experiences with the specific goal of achieving improved health and wellbeing.” This acknowledges that nature-based experiences can confer a range of benefits beyond physical activity promotion. This is better suited to come earlier on when showcasing the range of health benefits of NBHI before narrowing the focus to physical activity.
Introduction:
I would delete ‘ also’ in the following sentence because it already says ‘additionally’ at the start: “Additionally, NBHI have also been shown to further reduce costs on 50 healthcare systems, therefore there are multiple benefits of increasing and innovating NBHI”
Introduction:
The Introduction would be enhanced by expanding upon the research gap. Section 1.2 on Page 2 is quite brief. It could be a bit more fleshed out and incorporate relevant evidence highlighting a gap in knowledge to provide a greater justification for the need for this study and its contribution to the literature.
Materials and Methods:
I think there is a typo on page 2 in this sentence: ‘Step 3 included analyzing descriptive 71 statics and step 4, qualitative analysis including open coding of themes.” Should it be ‘statistics’ instead of ‘statics’?
Material and methods:
On page 4 section 2.4, the authors discuss the qualitative data analysis approach. The tool used to perform the analysis should be noted. Were the data analysed in Nvivo for example?
Materials and Methods:
In the first sentence in this section, I would refer to this study as a pilot study. This is not made clear until section 3 (Results) in the first sentence in 3.1.
Results:
In Table 1, one of the ‘N’ numbers is misaligned for ‘slightly familiar’ for the component, Familiarity with NBHI.
Results:
In section 3.2, add ‘of’ after ‘weaknesses’ and before ‘NBHI/prescriptions’ in the first sentence: The qualitative analysis included thematic analysis of open-ended questionnaire 192 responses about strengths and weaknesses NBHI/prescriptions.
Results:
The authors state that there were 4 themes in the following sentence on page 8 but then five are listed: “Thematic analysis of qualitative, open-ended questionnaire responses about 218 weaknesses of NBHI/prescriptions resulted in four themes including 1) trust and 219 acceptance, 2) support and resources, 3) implementation challenges, 4) access and 220 accessibility, and 5) environmental concerns.”
Comments on the Quality of English Language
See this discussed in general comments above.
Author Response
Response to Reviewer 1 Comments
Dear reviewer 1,
Thank you for your helpful time and consideration in helping us enhance our manuscript with further clarity and insights. We’ve addressed each of your comments within the manuscript in track changes, highlighted changes in yellow (attached) and orange (below), and provided detail about each edit below:
ABSTRACT
Comment 1: I would add an ‘and’ between ‘(74%)’ and ‘moderately’ and delete the semicolon separating these two clauses in the following sentence: “Qualitative themes 18 included strengths and weaknesses of NBHI. Participants (n= 53) were familiar with terms nature 19 play (82%), forest bathing (78%), park and nature prescriptions (74%); moderately familiar with 20 NBHI (5-point Likert scale, M = 3.27, SD = 1.17).”
Response 1: Thank you for your helpful correction. We added an ‘and’ between ‘(74%)’ and ‘moderately’ and deleted the semicolon separating these two clauses in the following sentence: “Qualitative themes 18 included strengths and weaknesses of NBHI. Participants (n= 53) were familiar with terms nature 19 play (82%), forest bathing (78%), park and nature prescriptions (74%) and moderately familiar with 20 NBHI (5-point Likert scale, M = 3.27, SD = 1.17).”
INTRODUCTION
Comment 2: Currently, the Introduction section goes straight into section 1.1, which clearly has a strong focus on physical activity in nature. I think that the Introduction could be strengthened by having a brief section for 1.0 before going into 1.1. In this additional part, this is where I would set the scene with what we know about the importance of exposure to nature and the different experiences (e.g., mediation, socialising, physical activity) that can lead to broader health benefits, such as stress reduction, mood improvement, anxiety reduction, social connection, sense of place, sense of community, active living. Then, from there you can go into the specific focus on physical activity. The reason for this suggested change is that the title of the article gives the impression that the focus is on the range of health benefits of NBHI, but right from the start, the Introduction mainly just focuses on physical activity in nature. Further, on page 2 of the Introduction, there is the following sentence, “NBHI are programs, activities, or strategies that aim to engage people in nature–based experiences with the specific goal of achieving improved health and wellbeing.” This acknowledges that nature-based experiences can confer a range of benefits beyond physical activity promotion. This is better suited to come earlier on when showcasing the range of health benefits of NBHI before narrowing the focus to physical activity.
Response 2: Thank you for your helpful insights. We agree that the Introduction could be strengthened by having a brief section for 1.0 before going into 1.1 setting the scene with what we know about the importance of exposure to nature and the different experiences including mediation, socialising, physical activity. We include that this exposure and experiences can lead to broader health benefits including stress reduction, mood improvement, anxiety reduction, social connection, sense of place, sense of community, active living before specifically focusing on physical activity. Additionally, we move the sentence you identify on page 2 of the Introduction to the beginning of the section, “NBHI are programs, activities, or strategies that aim to engage people in nature–based experiences with the specific goal of achieving improved health and wellbeing.” To further acknowledge that nature-based experiences can confer a range of benefits of NBHI before narrowing the focus to physical activity.
We include:
“It is clear that exposure to nature provides important and vast benefits from vitamin D absorption, additional immune response, and stress reduction1,2 to mediation, socializing, and physical activity3,4. These benefits can lead to a sense of life satisfaction and happiness due to mood improvement, anxiety reduction, social connection, sense of place, sense of community, and active living5. Naure Based Health Interventions (NBHI) are programs, activities, or strategies that aim to engage people in nature–based experiences with the specific goal of achieving improved health and wellbeing4,5. Additionally, NBHI have been shown to further reduce costs on healthcare systems2,5, therefore there are multiple benefits of increasing and innovating NBHI6."
Comment 3: I would delete ‘ also’ in the following sentence because it already says ‘additionally’ at the start: “Additionally, NBHI have also been shown to further reduce costs on 50 healthcare systems, therefore there are multiple benefits of increasing and innovating NBHI”.
Response 3: Thank you for your helpful suggestion. We agree this enhances clarity and delete ‘ also’ in the following sentence: “Additionally, NBHI have been shown to further reduce costs on 50 healthcare systems, therefore there are multiple benefits of increasing and innovating NBHI”.
Comment 4: The Introduction would be enhanced by expanding upon the research gap. Section 1.2 on Page 2 is quite brief. It could be a bit more fleshed out and incorporate relevant evidence highlighting a gap in knowledge to provide a greater justification for the need for this study and its contribution to the literature.
Response 4: Thank you for your insight. We agree the Introduction would be enhanced by expanding upon the research gap fleshing it out and incorporating relevant evidence highlighting a gap in knowledge to provide a greater justification for the need for this study and its contribution to the literature.
We include:
“There are varying perceptions about NBHI, uses of different terminology, and opinions about how stakeholders participate with NBHIs3. Additionally, the measurement of and operationalization of these interventions is vast and often limited4. It is unclear if an operationalized framework or measures could be understood among the may fields of stakeholders. Additionally, the roles these stakeholders are open to serving in and currently participating in is uncertain. The purpose of this mixed methods, cross-sectional study was to better understand perceptions, terminology, and participation in NBHI to guide future research priorities.”
METHODS
Comment 5: I think there is a typo on page 2 in this sentence: ‘Step 3 included analyzing descriptive 71 statics and step 4, qualitative analysis including open coding of themes.” Should it be ‘statistics’ instead of ‘statics’?
Response 5: Thank you for pointing out our oversight. We have corrected the sentence on page 2 to be ‘statistics’ instead of ‘statics’?: ‘Step 3 included analyzing descriptive statistics and step 4, qualitative analysis including open coding of themes.”
Comment 6: On page 4 section 2.4, the authors discuss the qualitative data analysis approach. The tool used to perform the analysis should be noted. Were the data analysed in Nvivo for example?
Response 6: Thank you for this opportunity to provide this detail. On page 4 section 2.4, we include the tool used to perform the analysis was Microsolt Excel.
Comment 7: In the first sentence in this section, I would refer to this study as a pilot study. This is not made clear until section 3 (Results) in the first sentence in 3.1.
Response 7: Thank you for your recommendation. In the first sentence in this section, we now would refer to this study as a pilot study.
RESULTS
Comment 8: In Table 1, one of the ‘N’ numbers is misaligned for ‘slightly familiar’ for the component, Familiarity with NBHI.
Response 8: Thank you for finding our error. In Table 1, we corrected the ‘N’ number that was misaligned for ‘slightly familiar’ for the component, Familiarity with NBHI.
Comment 9: In section 3.2, add ‘of’ after ‘weaknesses’ and before ‘NBHI/prescriptions’ in the first sentence: The qualitative analysis included thematic analysis of open-ended questionnaire responses about strengths and weaknesses NBHI/prescriptions.
Response 9: Thank you for your recommendation. In section 3.2, we added ‘of’ after ‘weaknesses’ and before ‘NBHI/prescriptions’ in the first sentence: “The qualitative analysis included thematic analysis of open-ended questionnaire responses about strengths and weaknesses of NBHI/prescriptions.”
Comment 10: The authors state that there were 4 themes in the following sentence on page 8 but then five are listed: “Thematic analysis of qualitative, open-ended questionnaire responses about 218 weaknesses of NBHI/prescriptions resulted in four themes including 1) trust and 219 acceptance, 2) support and resources, 3) implementation challenges, 4) access and 220 accessibility, and 5) environmental concerns.”
Response 10: Thank you for finding our error. We correct that there were five themes in the following sentence on page 8: “Thematic analysis of qualitative, open-ended questionnaire responses about weaknesses of NBHI/prescriptions resulted in five themes including 1) trust and 219 acceptance, 2) support and resources, 3) implementation challenges, 4) access and 220 accessibility, and 5) environmental concerns.”

Reviewer 2 Report
Comments and Suggestions for Authors
The topic of nature-based health interventions is highly relevant, especially given the growing interest in the intersection of nature and health. The study’s focus on perceptions, terminology, and participation in NBHI fills an important gap in understanding how these interventions are viewed by stakeholders. This research adds value by addressing both the familiarity with NBHI and the potential for its broader application, which is crucial for public health promotion.
It's unclear why the introduction starts with the link between nature-based interventions and physical activity; it should begin with a more conceptual and general overview before narrowing down to specifics. If the focus is on physical activity, then the title of the article should be adjusted accordingly.
In my perspective, the use of a mixed-methods, cross-sectional design is appropriate for exploring the complex perceptions and experiences related to NBHI. The results are presented clearly, with specific statistics (e.g., familiarity percentages, Likert scale scores) that provide a snapshot of the findings.
Several questions arise when discussing nature-based health interventions:
· Are there specific types of nature settings that are more beneficial for certain health outcomes?
· How can NBIs be made accessible to diverse populations, including those in urban, rural, or underserved communities?
· What are the barriers to participation in NBIs, and how can they be addressed?
· How do cultural perceptions of nature influence participation in and the effectiveness of NBIs?
· Should NBIs be designed to also address climate-related health risks, such as heatwaves or pollution?
· How can NBIs contribute to climate resilience and environmental sustainability?
It may not be possible to address all these points in detail, but the discussion could briefly touch on these topics to encourage reflection.
I believe that the recommendations to "operationalize components to increase use" could be elaborated upon slightly, to give a sense of what specific actions or policies might be needed. Also, please include 2 or 3 studies from 2023-2024 in this area to update the current information available on the topic.
Author Response
Response to Reviewer 2 Comments
Dear reviewer 2,
Thank you for your helpful time and consideration in helping us enhance our manuscript with further clarity and insights. We have addressed each of your comments within the manuscript in track changes, highlighted changes in yellow, and provided detail about each edit below:
INTRODUCTION
Comment 1: It's unclear why the introduction starts with the link between nature-based interventions and physical activity; it should begin with a more conceptual and general overview before narrowing down to specifics. If the focus is on physical activity, then the title of the article should be adjusted accordingly.
Response 2: Thank you for your helpful insights. We agree that the introduction could be strengthened by begin with a more conceptual and general overview before narrowing down to specifics. Additionally, we move the sentence on page 2 of the introduction to the beginning of the section, “NBHI are programs, activities, or strategies that aim to engage people in nature–based experiences with the specific goal of achieving improved health and wellbeing.” to further acknowledge that nature-based experiences can confer a range of benefits of NBHI before narrowing the focus to physical activity.
We include:
“It is clear that exposure to nature provides important and vast benefits from vitamin D absorption, additional immune response, and stress reduction1,2 to mediation, socializing, and physical activity3,4. These benefits can lead to a sense of life satisfaction and happiness due to mood improvement, anxiety reduction, social connection, sense of place, sense of community, and active living5. Nature Based Health Interventions (NBHI) are programs, activities, or strategies that aim to engage people in nature–based experiences with the specific goal of achieving improved health and wellbeing4,5. Additionally, NBHI have been shown to further reduce costs on healthcare systems2,5, therefore there are multiple benefits of increasing and innovating NBHI6.”
METHODS AND RESULTS
Comment 2: In my perspective, the use of a mixed-methods, cross-sectional design is appropriate for exploring the complex perceptions and experiences related to NBHI. The results are presented clearly, with specific statistics (e.g., familiarity percentages, Likert scale scores) that provide a snapshot of the findings.
Response 2: Thank you for your time and expertise in reviewing these sections.
DISCUSSION
Comment 3: Several questions arise when discussing nature-based health interventions:
- Are there specific types of nature settings that are more beneficial for certain health outcomes?
- How can NBIs be made accessible to diverse populations, including those in urban, rural, or underserved communities?
- What are the barriers to participation in NBIs, and how can they be addressed?
- How do cultural perceptions of nature influence participation in and the effectiveness of NBIs?
- Should NBIs be designed to also address climate-related health risks, such as heatwaves or pollution?
- How can NBIs contribute to climate resilience and environmental sustainability?
It may not be possible to address all these points in detail, but the discussion could briefly touch on these topics to encourage reflection.
Response 3: Thank you for your insights. We agree the discussion would be enhanced by expanding upon the points you raise. We touch on these topics to encourage reflection as noted:
We expand by including:
“Much work is needed to better understand what the operationalized framework we recommend might look like that addresses access, participation barriers, and cultural perceptions between diverse populations at each level of the SME. It is suggested to focus on human-nature as a two-way, beneficial relationship that might reduce barriers and improve sustainability33. However, is often left out in health and conservation policies33. One suggested framework model that might uncover these pathways and capture some of the differences between cultures and populations is "A time with e-Natureza" (e-Nature) introduced by Leão et al (2023)34. This framework focuses on the underlying interactions of nature-based health interventions including (1) Aesthetic and emotional experience; (2) Multisensory integration experience; (3) Knowledge experience; and (4) Engagement experience. It seems this model would need further development to fully consider climate-related health risks, such as heatwaves or pollution though it seems the “multisensory integration experience” may be used to consider this area. Therefore, limited information is understood about the contribution to climate resilience and environmental sustainability (Barragan-Jason et al., 2023) and current research is also unclear if types of nature settings are more beneficial for certain health outcomes. Use of validated instruments to measure attitudes35, self-efficacy and intentions about spending time in nature36 as well as the setting itself such instruments as NatureScore may assist in capturing a more comprehensive benefits of nature on health37.”
Comment 5: I believe that the recommendations to "operationalize components to increase use" could be elaborated upon slightly, to give a sense of what specific actions or policies might be needed. Also, please include 2 or 3 studies from 2023-2024 in this area to update the current information available on the topic.
Response 5: Thank you for pointing out this opportunity to elaborate on specific actions or policies that might be needed to "operationalize components to increase use". We agree this elaboration would enhance our manuscript. Please see this content integrated in the content for response 4.
We include 3 studies from 2023-2024 along with two from 2022 in this area to update the current information available on the topic:
(33) Barragan-Jason G, Loreau M, de Mazancourt C, Singer MC, Parmesan C. Psychological and physical connections with nature improve both human well-being and nature conservation: A systematic review of meta-analyses. Biological Conservation. 2023;277:N.PAG. doi:10.1016/j.biocon.2022.109842
(34) Leão ER, Hingst-Zaher E, Savieto RM, et al. A time with e-Natureza (e-Nature): a model of nature- based health interventions as a complex adaptive system. Front Psychol. 2023;14:1226197. Published 2023 Aug 22. doi:10.3389/fpsyg.2023.1226197
(35) Maddock JE, Suess C, Bratman GN, et al. Development and Validation of an Attitude Toward Spending Time in Nature Scale. Ecopsychology. 2022;14(3):200-211. doi:10.1089/eco.2022.0017
(36) Maddock JE, Suess C, Bratman GN, et al. Development and validation of self-efficacy and intention measures for spending time in nature. BMC Psychology. 2022;10(1). doi:10.1186/s40359-022-00764-1
(37) Makram OM, Pan A, Maddock JE, Kash BA. Nature and Mental Health in Urban Texas: A NatureScore-Based Study. International Journal of Environmental Research and Public Health. 2024; 21(2):168. https://doi.org/10.3390/ijerph21020168
